# Natural and Artificial Sweeteners and High Fat Diet Modify Differential Taste Receptors, Insulin, and TLR4-Mediated Inflammatory Pathways in Adipose Tissues of Rats

**DOI:** 10.3390/nu11040880

**Published:** 2019-04-19

**Authors:** Mónica Sánchez-Tapia, Jonathan Martínez-Medina, Armando R Tovar, Nimbe Torres

**Affiliations:** Departamento de Fisiología de la Nutrición, Instituto Nacional de Ciencias Médicas y Nutrición Salvador Zubirán, México City 14080, Mexico; qfbmoniktc@gmail.com (M.S.-T.); jonathan.martinez@comunidad.unam.mx (J.M.-M.); armando.tovarp@incmnsz.mx (A.R.T.)

**Keywords:** adipose tissue, sweeteners, taste receptors, TLR4, BCAA, insulin signaling, incretins

## Abstract

It is difficult to know if the cause for obesity is the type of sweetener, high fat (HF) content, or the combination of sweetener and fat. The purpose of the present work was to study different types of sweeteners; in particular, steviol glycosides (SG), glucose, fructose, sucrose, brown sugar, honey, SG + sucrose (SV), and sucralose on the functionality of the adipocyte. Male Wistar rats were fed for four months with different sweeteners or sweetener with HF added. Taste receptors T1R2 and T1R3 were differentially expressed in the tongue and intestine by sweeteners and HF. The combination of fat and sweetener showed an additive effect on circulating levels of GIP and GLP-1 except for honey, SG, and brown sugar. In adipose tissue, sucrose and sucralose stimulated TLR4, and c-Jun N-terminal (JNK). The combination of HF with sweeteners increased NFκB, with the exception of SG and honey. Honey kept the insulin signaling pathway active and the smallest adipocytes in white (WAT) and brown (BAT) adipose tissue and the highest expression of adiponectin, PPARγ, and UCP-1 in BAT. The addition of HF reduced mitochondrial branched-chain amino transferase (BCAT2) branched-chain keto acid dehydrogenase E1 (BCKDH) and increased branched chain amino acids (BCAA) levels by sucrose and sucralose. Our data suggests that the consumption of particular honey maintained functional adipocytes despite the consumption of a HF diet.

## 1. Introduction

In the last few decades there has been an increase in the consumption of natural and artificial sweeteners [1]. The increase in the consumption of natural sweeteners, particularly sucrose (sugar) has been associated with the obesity epidemic worldwide [2]. One of the strategies in reducing this health problem has been the use of artificial sweeteners to reduce energy intake. However, in the recent years, controversial results have been reported about the consumption of artificial sweeteners that have been associated with a greater risk of being overweight or obese [3]. Besides that, other studies have focused on other monosaccharides, particularly fructose [4], and few studies have focused on other natural complex sweeteners, such as honey, brown sugar, steviol glycosides, or the artificial sweetener sucralose [5]. In conditions of persistent intake of sweeteners, some taste receptors type 1 member 2 (T1R2) or member 3 (T1R3) are activated in tongue, intestine, and adipose tissue [6,7,8]. Sugar and artificial sweeteners can bind selectively to one or both receptors in different proportions, and this raises the possibility that sweeteners may influence the metabolism of adipose tissue [9]. Adipose tissue is a dynamic endocrine organ that responds to energy intake by increasing its size and/or number of adipocytes to facilitate lipid storage. In the intestine, sweeteners can activate taste receptors regulating the secretion of the incretins glucose-dependent insulinotropic peptide (GIP) and glucagon-like peptide-1 (GLP-1) [10]. The GIP receptor (GIPr) is present in white and brown adipose tissue [11] and regulates insulin action [12] and pro-inflammatory cytokine production [13] leading to changes in adipocyte functionality. Many adipose nutrient-sensing pathways are involved in the regulation of adipose tissue metabolism. Among these are, hormones, such as insulin signaling [14], or by regulating the secretion of adiponectin, leptin and other cytokines [15]. Another possible mechanism, for which a sweetener could regulate the functionality of the adipocyte, is through toll like receptors (TLR) that, in turn, regulate several inflammatory pathways in the adipocyte [16]. However, the effect of natural and artificial sweeteners, on the function of adipocytes, has been not fully studied. In addition, the presence of high fat in the diet in combination with the different sweeteners could modify the different metabolic pathways involved in the functionality of the adipocyte. Thus, the purpose of the present work was to study the effect of monosaccharides, disaccharides, natural complex sweeteners, particularly brown sugar, honey, steviol glycosides, and the artificial sweetener sucralose on the tongue and intestine taste receptors and its role in incretins, insulin signaling, and potentially TLR 4 pathway, and branched chain amino acids in adipose tissue in the presence or absence of a high fat diet to better understand potential molecular mechanism (s), on which sweeteners modulate adipose metabolism.

## 2. Materials and Methods

### 2.1. Animals and Diets

Male Wistar rats aged 5 weeks were obtained from the National Institute of Medical Sciences and Nutrition. The animals were housed in individual cages and maintained at a controlled room temperature with 12-h light-dark cycles and free access to water and food. Rats were randomized into 18 groups; nine groups were fed a control diet (C) according with AIN-93 [17] and different sweeteners in drinking water (W) (*n* = 6 per group); the other 9 groups were fed high-fat diet (HF) and different sweeteners in drinking water at the following concentration: Sucrose (S), fructose (F), glucose (G), brown sugar (BS), honey (H), steviol glycosides, plus sucrose (SV); at 10%, steviol glycosides (SG) at 2.5% and sucralose (SU) at 1.5%, for four months (*n* = 6 per group). The high fat diet (HF) consisted in 17% lard (45% kcal from fat). A group of 24 rats were fed with control (C) diet (*n* = 12) or high fat diet (*n* = 12) to study the effect of antibiotics on branched chain amino acids (BCAA). Six rats of the control or HF group received antibiotic (Cas-Abx or HF-Abx) and the rest of the rats continued consuming the C or HF diet. Antibiotics were provided in the drinking water for 3 weeks and the concentration of antibiotics were 1 mg/mL ampicillin and 0.5 mg/mL neomycin. At the end of study, the rats were anesthetized with sevoflurane before decapitation. Inguinal white adipose tissue and interscapular brown adipose tissue, tongue, small intestine and colon were rapidly removed and stored at −70 °C. Serum was obtained by centrifugation of blood at 1500× *g* for 10 min and stored at −70 °C. The Animal Committee of the National Institute of Medical Sciences and Nutrition, Mexico City approved the procedure (CINVA1735).

### 2.2. Biochemical Parameters

Serum glucose was analyzed with the autoanalyzer COBAS C11 (Roche, Basel, Switzerland). Insulin by ELISA kit (Alpco Diagnostics, Salem, NH, USA), leptin was determined using commercial ELISA kits (Abcam, Cambridge, MA, USA).

### 2.3. White and Brown Adipose Tissue Gene Expression

Gene expression was determined by real-time PCR. First, the total RNA was extracted using TRIzol, following the manufacturer’s instructions. The mRNA abundance was measured by real-time quantitative PCR using SYBR^®^ Green assays (Roche), using 36B4 and cyclophilin as housekeeping gene for normalization. All primer sequences for gene expression determination are shown in Appendix A.

### 2.4. Western Blot Analysis

The total protein of pooled adipose, tongue, small intestine, and colon tissues samples (*n* = 6 for each group) was extracted, quantified by Bradford assay (Bio-Rad, Hercules, CA, USA), and stored at −70 °C. The protein detection was performed by electrophoresis in SDS-PAGE and then transferred to polyvinylidene difluoride membranes. All blots were blocked with 3% BSA for 60 min at room temperature and incubated overnight at 4 °C with primary antibodies. For adipose tissue, the following antibodies were used: Toll-like receptor 4 (TLR-4) (1:1000), nuclear factor-kappa B (NF-κB) (1:1000), kinase c-Jun N-terminal (JNK) (1:1000), myeloid differentiation primary response 88 (MyD 88) (1:1000), total insulin receptor substrate (IRS) (1:750), phosphorylated IRS (p263y-IRS) (1:2000), total protein kinase B (AKT)(1:2500) and phosphorylated AKT(1:5000). For tongue and colon, the following primary antibodies were used: Taste 1 receptor member 2 (T1R2) (1:1000) and taste 1 receptor member 3 (T1R3) (1:1000). For small intestine, the primary antibodies used were sodium-glucose co transporter type 1 (SGLT-1) (1:1500), glucose transporter 2 (GLUT2) (1:1000), glucagon-like peptide-1 (GLP-1) (1:3500), gastric insulinotropic peptide (1:2500), T1R2 and T1R3. The blots were incubated with anti-rabbit, anti-goat, or anti-mouse secondary antibodies conjugated with horseradish peroxidase (1:15000). GAPDH (1:3500) was used to normalize the data. Images were analyzed with a ChemiDocTM XRS + System Image LabTM Software (Bio-Rad, Hercules, CA, USA). The western blot analysis was performed three times using independent blots.

### 2.5. Histological Analysis

Adipose tissue sections 4 μm thick were stained with hematoxylin and eosin. A morphological analysis was performed using a Leica microscope (Leica DM750 Wetzlar, Germany). Adipocyte size measurement was performed using AdipoSoft (ImageJ, 1.14 version, NIH, Bethesda, MD, USA).

### 2.6. Statistical Analysis

The results were expressed as the mean ± SEM. Statistical analysis was performed using one-way ANOVA followed by Bonferroni’s post-hoc test, using Prism 7.0 software (GraphPad, San Diego, CA, USA); *p* < 0.005 was considered significant.

## 3. Results

### 3.1. Body Weight and Biochemical Parameters

After 4 months of dietary treatment with sweeteners, rats fed with sucrose, SV, or sucralose gained significantly more weight than those fed brown sugar, glucose, or honey, whereas those fed with SG, fructose or no sweetener in the water had the lowest weight gain (*p* = 0.04). The presence of HF in the diet significantly increased body weight in most of the groups, except for those fed with sucralose or SG. Similar to body weight, rats fed with sucralose, sucrose, or SV had the highest serum glucose concentration than the rest of the groups. Those fed with glucose, fructose or brown sugar had a higher serum glucose than those fed SG, honey or without sweetener. The addition of high fat in the diet increased serum glucose in rats fed sucralose, sucrose, SG, fructose or no sweetener. It is noteworthy, that rats fed with honey had the lowest serum glucose (*p* < 0.001). Interestingly, long term consumption of sucralose produced hyperinsulinemia followed by sucrose or SV, whereas those fed brown sugar, honey, SG, or no sweetener had the lowest serum insulin. Insulin concentration in rats fed sweeteners plus HF in the diet followed the same pattern than without HF. Unexpectedly, honey was the only sweetener that, even in the presence of HF, did not significantly increase insulin concentration. Although rats fed sucralose, sucrose and SV had similar body weight, and those fed with sucralose developed hyperleptinemia. Rats that were fed BS, honey, or no sweetener showed the lowest serum leptin concentration. The presence of HF in the diet significantly increased serum leptin concentrations in all groups, however, those fed sucralose, sucrose, SV, glucose, fructose, or no sweetener developed hyperleptinemia, Table 1.

### 3.2. Tongue and Intestine TIR2, TIR3, SGLT-1, GLUT-2 and Incretins After the Consumption of Natural and Artificial Sweeteners With and without a High Fat Diet

Tongue taste receptors T1R2 and T1R3 serve as sensors of different types of sweeteners. TIR2 was significantly induced by SG, glucose, sucrose, honey, brown sugar, SV and sucralose by 1.53-, 1.2-, 1.02-, 0.94-, 0.83-, 0.66-, and 0.57-fold, respectively compared to the control group (C + W). Interestingly, the presence of high fat in the diet reduced in a higher extent the abundance of T1R2, Figure 1A. T1R3 was significantly more stimulated by sucrose than T1R2. T1R3 significantly stimulated sucrose, fructose, brown sugar, and sucralose by 2.5-, 1.5-, 1.3-, and 1.2-fold, respectively compared to the C+W group. Also, T1R3 was reduced by the presence of HF in the diet with the exception of glucose and SG, Figure 1A.

TIR2 and TIR3 in the intestine may play a role in post-oral absorption and metabolism of sweeteners and could stimulate the secretion of incretins, that in turn regulate the glucose transporters and insulin secretion [18]. The abundance of T1R2 in small intestine and colon followed a similar pattern. Sucralose and SG stimulated the abundance of T1R2 to a higher extent than the rest of the sweeteners, followed by glucose, fructose, and sucrose, and to a lesser extent by BS and H, Figure 1B. Sucralose, SV and sucrose significantly induced the abundance of T1R3. On the contrary to the tongue, T1R2 and T1R3 abundance was stimulated by the presence of HF in the diet in small intestine and mainly in the colon, Figure 1C. Interestingly, sodium-dependent glucose cotransporter 1 (SGLT-1) abundance, which is found in the enterocytes of the small intestine, was significantly stimulated by glucose, fructose, BS, sucrose, H, sucralose, SG and SV by 4.6-, 3.3-, 2.5-, 2.0-, 1.6-, 1.5-, 1.2- and 0.9-fold respectively compared to the control group. The presence of high fat in the diet significantly increased the abundance of SGLT-1 by glucose, fructose, BS, sucralose, and H by 5.7-, 4.1-, 3.2-, 2.7-, and 2.2-fold, respectively, compared to the control group. The glucose transporter 2 (GLUT2), which facilitates the movement of glucose across cell membrane, was induced by all sweeteners with the lowest abundance by SG, however the presence of a high fat diet increased its abundance by all the sweeteners with the lowest induction by SG, Figure 1D.

Sweeteneers specifically stimulate taste receptors, that in turn can regulate the release of the incretins glucose insulino tropic peptide (GIP), and glucagon like peptide (GLP-1) [19]. GLP-1 in the small intestine was induced by sucrose and sucralose by approximately 4.1-, and 3.5-fold, respectively, compared to the control group, whereas only the addition of high fat to the diet whithout sweeteners, increased the abundance of GLP-1 by 2.2-fold in comparison with the C + W group. the consumption of a high fat with sucralose, SV, sucrose, glucose, fructose, and SG increased GLP-1 protein abundance significantly by 8.7-, 6.1-, 5.9-, 4.1-, 3.9-, and 3.2-fold, respectively. Interstingly, H showed no increase in GLP-1 abundance even in the presence of a high fat diet. GIP was induced by SV, sucralose, and sucrose by 15.4-, 11.4-, and 10.3-fold, and the presence of high fat in the diet induced, to a large extent, GIP abundance by sucrose, sucralose, and glucose by 21.1-, 14.9-, 12.5-fold, respectively, compared to the control group C+W, Figure 1E. These results were confirmed when these incretins were measured in serum. Sucralose and sucrose were the main stimulators of GIP and GLP-1 followed by fructose, glucose, SV, and finally by SG, BS, or H. The presence of high fat in the diet showed an additive effect on GIP mainly by sucralose and sucrose, indicating that no only the type of sweetener, but the presence of saturated fat play, an important role in the circulating levels of incretins, Figure 1E,F.

### 3.3. Inflammatory Pathways are Differentially Regulated by the Type of Sweeteners and High Fat

Adipocytes may play an important role in inflammation and insulin resistance via TLR signaling after the consumption of sweeteners. Rats fed with sucrose, sucralose, and glucose showed the highest expression of RNA and protein abundance of TLR4, followed by fructose, Figure 2A,B,F. The lowest expression of TLR4 was observed with BS and honey. It has been demonstrated that the interaction of TLR4 with MyD88 triggers a downstream signaling cascade, leading to the activation of the transcription factor NF-κB pathway and reciprocally inducing the activation of the c-Jun N-terminal kinase inducing a proinflammatory state [20]. In the present study, MyD88 followed a similar pattern than TLR4, Figure 2A,C. The highest protein abundance of MyD88 was observed with sucrose followed by sucralose. JNK was highly phosphorylated by SV, sucralose, sucrose, and fructose, and the addition of a high fat diet significantly increased p-JNK, with the exception of H, Figure 2A,D. Interestingly, there was no differential induction of NFκ protein abundance by any sweetener, however the presence of high fat in the diet, induced to a larger extent, the protein abundance of NFκ by sucralose, followed by sucrose, and there was no stimulation of NFκ by the groups fed SG and H, Figure 2A,E. These results indicated that both pathways, regulated by MyD 88, are differentially activated by sweeteners, and to a greater extent, by saturated fat. Finally, NFκB and JNK play an important role in the production of proinflammatory cytokines. In fact, cytokine TNF was stimulated mainly by sucralose, SV, sucrose, glucose, and fructose and in a lesser extent by SG, BS and H, and the addition of high fat significantly increased the gene expression of this cytokine.

### 3.4. Insulin Signaling after the Consumption of Sweeteners

Incretins, secreted from enteroendocrine cells, may play a critical role in the development of obesity by regulating insulin signaling in adipocytes, depending on the type of sweetener consumed. To further investigate the insulin-mimetic effects of GIP via activation of AKT and its effect on GLUT4, we studied the effect of the different sweeteners. Honey, BS, and SG showed adequate insulin signaling by maintaining the phosphorylation of IRS and AKT, however, sucralose and sucrose showed a significant decrease in the phosphorylation of pT263 ^IRS1^ and pS473^AKT^, Figure 3A–C. The addition of high fat in the diet decreased the phosphorylation of IRS and AKT with the exception of H and BS, as shown in Figure 3A–C. As a result, the expression of GLUT 4 was induced only by BS and H, as evident in Figure 3D. These results indicate that sweeteners specifically regulate insulin signaling and the presence of high fat in the diet significantly decreased the insulin signaling. On the other hand, the consumption of sucralose, and to a lesser extent, sucrose, produced hyperinsulinemia and significantly increased lipogenesis mediated by SREBP-1, Figure 3E.

### 3.5. The Type of Sweetener and High Fat Diet Differentially Modify the Functionality of the Adipocyte in White Adipose Tissue

The adipocyte size of rats fed control diet, without sweetener, was approximately 4743 µm^2^, and the consumption of a high fat doubled the size of the adipocytes. Interestingly, the consumption of specific sweetners had a significant impact on the size of the adipocytes. Rats fed sucrose and sucralose showed the highest size of adipocytes, approximately 7.5- and 6.3-fold higher compared to those fed the control diet, whereas the lowest size was seen in rats fed H and BS, Figure 4A,C. The addition of high fat in the diet had no significant effect on the size of the adipocyte, Figure 4B,C. Almost all sweeteners decreased the expression of PPARγ involved in adipogenesis, and with some sweeteners, the addition of a high fat diet further decreased the expression of this gene. Surprinsingly, this effect was not observed with BS, but was especially seen with honey, Figure 4D. Interestingly, the gene expression of adiponectin was significantly low in rats fed all the sweeteners with the exception of SG, H and BS that increased adiponectin mRNA abundance by 6.4-,10.5- and 4.3-fold respectively, however the addition of a high fat diet significantly reduced the expression of adiponectin with all sweeteners. Leptin is an adipokine, which has been shown to increase with increased fat mass, and our results showed that all sweeteners had a low impact on the leptin concentration, however, the presence of a high fat diet significantly increased leptin mRNA abundance with all sweeteners, mainly with sucralose that had a 2.6-fold increase with respect to the sweetener alone, followed by sucrose, fructose, SV, glucose, SG, and honey.

### 3.6. Sweeteeners Differentially Regulate Gene Expression of Adiponectin and UCP-1 in BAT

The size of the brown adipocytes in control rats without sweetener was around 124 µm^2^. Animals fed with sucrose (720 µm^2^), sucralose (630 µm^2^), or fructose (630 µm^2^) showed the biggest brown adipocyte size followed by glucose and SG, and the lowest size was observed for SV (150 µm^2^), brown sugar (261 µm^2^) and honey (283 µm^2^), (Figure 5A,B). Interestingly, the histological analysis revealed the presence of large lipid droplets in the groups of rats fed with sucralose, sucrose, fructose and glucose indicating a possible whitening of the adipocytes (Figure 5A). The addition of high fat content in the diet significantly increased the adipocyte size mainly in rats fed sucrose, glucose, SG and SV (Figure 5C,D). Similar to the WAT, only the groups fed with H, BS and SG showed the smallest size of adipocyte (Figure 5D). Also, as observed in WAT, the highest levels of adiponectin (10-fold) were observed in H, BS and SG (Figure 5E). Interestingly, rats fed with SG, BS, or honey significantly increased uncoupled protein- (UCP-1), a thermogenesis marker, however the presence of high fat in the diet significantly decreased UCP-1 mRNA abundance (Figure 5F).

### 3.7. High Fat Diet and Specific Sweetners Increased the Concentrations of Branched Chain Amino Acids

Adipose tissue can metabolize substantial amounts of branched chain amino acids, and an increase of these amino acids has been reported during obesity. For this reason, we measured serum branched chain amino acid concentration in rats fed different sweeteners, with, and without, the presence of a high fat diet. Animals fed the control diet had serum BCAA levels in the range of 400 ng/mL, whereas the consumption of sucrose or sucralose increased by approximately 3-fold BCAA concentration, and rats fed with SG, honey and brown sugar presented the similar levels than the control group. The addition of only fat in the diet significantly increased by 1.9-fold the BCAA concentration and the combination of sucrose or sucralose with HF increased by 5.6-, and 5.5-fold the concentration of these amino acids, respectively. In the rest of the groups, the presence of fat in the diet had an additive effect on serum BCAA concentration, Figure 6E. To explain the increase of these amino acids, we measured the gene expression of the enzymes involved in the catabolism of BCAA, BCAT, and BCKD, in adipose tissue. The results showed similar expression of both enzymes in WAT (Figure 6A,B) and BAT (Figure 6C,D) with all sweeteners, but significantly decreased their expression in the presence of a high fat diet independently of the type of sweetener. However, the decrease in the expression of BCAT and BCKDH did not explain the elevation of BCAA concentration in serum. Thus, a group of rats was fed a control diet and the other group was fed a high fat diet, and after 5 weeks of dietary treatment, rats were given antibiotics in the drinking water for one month. This treatment produced a significant decrease in the levels of BCAA by 64% in the control group and by 55% in the high fat group (Figure 6F) indicating that the gut microbiota produced a significant amount BCAA.

## 4. Discussion

With the current obesity problems, new sweeteners with low calories have been incorporated; this situation has created confusion for the consumer about what type of sweetener is the best option. In addition to this, the high consumption of saturated fats has been associated with an increased risk of coronary heart disease. Thus, it is difficult to know if the responsible for obesity is the type of sweetener, the high saturated fat or the combination of sweetener and fat [21].

Our results demonstrated that persistent consumption of sweeteners, particularly the disaccharide sucrose or the artificial sweetener sucralose, increased the abundance of T1R2 and T1R3 taste receptors in tongue and small intestinal epithelium, and they function as luminal sugar sensors to control SGLT1 expression in response to dietary sugars [22]. It is important to point out that even low sucralose concentration (1.5%), used in this study, showed similar effects than 10% sucrose. These results could be in part due to the fact that sucralose is metabolized in rats to compounds that are less polar and more lipophilic than the parent compound and persisted in adipose tissue two weeks after cessation of use [5]. L cells of the gut contain taste receptors that modulate GLP1 secretion in the gut and may play an important role in the development of obesity, diabetes, and changes in gut motility [19]. We found that long-term consumption of sucrose or sucralose significantly increased the secretion of the incretins GIP and GLP-1 by 5.6-, 7.7-fold, leading to the highest body weight gain, hyperglycemia, hyperinsulinemia, and hyperleptinemia. In addition, rats fed with sucrose or sucralose also increased the abundance of GLUT 2 in the enterocytes, leading to an increase in the transport of glucose and the highest circulating blood glucose levels. An increase in GLUT2 has been demonstrated in human enterocytes of obese subjects, that are associated with insulin resistance and hyperglycemia [23]. Furthermore, the combination of sucrose or sucralose with a high fat diet aggravated all abnormalities indicated above.

Although additional studies will be necessary in humans to fully characterize the complex nature and the biological significance of each sweetener, the present work demonstrated that honey has a protective effect in the development of obesity, since rats that are fed honey, showed the lowest levels of incretins, insulin, serum glucose, and leptin and the lowest stimulation of inflammatory signaling pathways.

In addition to its role as the primary mediator of the enteroinsular axis, GIP may play a critical role in the metabolism of the adipocyte [24]. The elevation of GIP by sucrose or sucralose inactivate the phosphorylation of IRS_tyr_ and Akt, indicative of insulin resistance, condition that was aggravated by the presence of high fat diet. Interestingly, honey was the only sweetener that increased the phosphorylation of IRS_tyr_ and Akt, indicative of a better insulin signaling associated with the lowest concentration of blood glucose and insulin.

GLUT 4 is the insulin-sensitive glucose transporter which its main role is to provide glucose to the adipose tissue. Transgenic mice lacking GLUT 4, decrease whole-body insulin sensitivity [25]. The inactivation of Akt, can decrease the membrane translocation of GLUT4, leading to a reduced adipocyte glucose uptake resulting in hyperglycemia, effect mainly observed in the groups fed with sucralose and sucrose. Interestingly, only rats fed honey or brown sugar significantly increased white adipose GLUT 4 expression, even with the addition or high fat, indicative of a better insulin sensitivity.

Dysregulation of signaling pathways in adipose tissue can contribute to the development of obesity. In fact, we observed that the area of the adipocytes was significantly larger in rats fed with sucrose and sucralose with and without high fat in the diet. The increase in adipocyte size in the group fed with sucralose and sucrose was aggravated by stimulation of lipogenesis induced by the hyperinsulinemia via SREBP-1, mainly by sucralose. Remarkably, the group fed with honey showed the smallest adipocytes size and reduced lipogenesis even in the presence of high fat in the diet that was associated with the highest expression of PPARγ and adiponectin, indicative of an adequate adipocyte differentiation. Enlarged adipocytes recruit macrophages and promote inflammation and the release of factors that produce insulin resistance. The TLR4 signaling pathways are the main triggers of the obesity-induced inflammatory response [26]. Studies indicate that saturated fatty acid can induce inflammation by activating the TLR4 signaling pathway [26,27]. Sucralose and sucrose with and without high fat diet increased the abundance of this receptor, indicative of a stimulation of inflammatory pathways. Honey and SG showed the lowest abundance of TLR4. MyD 88 was increased by sucrose and sucralose independently of the presence of fat in the diet. The cJun-N-terminal-kinase has been associated with insulin resistance and the activation of the transcription factor NFκB. In this study, rats fed with sucrose or sucralose increased the phosphorylation of JNK with, and without, fat in the diet; however, this effect was not observed with honey only. It is noteworthy that only rats fed a high fat diet increased the protein abundance of NFκB, with the exception of the groups fed honey or SG. NFκB, in turn, stimulated proinflammatory cytokine production, and in particular, TNFα. Interestingly, only the consumption of a high fat diet stimulated gene expression of TNFα, and the combination of high fat plus any sweetener further increased the expression of TNFα. An increase in NFκB inhibits PPARγ function, this in turn inhibits adipogenesis promoting hypertrophy of adipocytes, the effect was observed in rats fed sucrose or sucralose. BS and particularly honey showed the opposite effect, maintaining insulin sensitivity, even in rats fed high fat diet.

On the other hand, insulin resistance has been associated with high levels of BCAA in humans [28], and in this study, the groups fed sucrose or sucralose showed the highest concentration of serum BCAA whereas those fed honey, SG and BS had the lowest concentration. Interestingly, treatment with antibiotics significantly reduced the circulating levels of BCAA, which is indicative of the importance of gut microbiota in the synthesis of BCAA. Finally, honey and BS maintained the phenotypic characteristics of brown adipose tissue, with a significant increase in UCP1. The present work demonstrated that honey has a protective effect in the development of obesity, maintaining insulin sensitivity, and low levels of serum glucose and insulin. In addition, the consumption of honey increased the functionality of the adipocyte by increasing adiponectin and PPARγ in WAT, UCP-1 in BAT, an indicator of thermogenesis, and reducing the inflammatory state mediated by TLR4. These findings found in honey may be due in part to the particular composition of the honey. Honey only contains 1% sucrose and other monosaccharides, such as glucose and fructose, around 150 polyphenolic compounds with antioxidant capacity [29,30] that could be able to maintain the adipocyte functionality.

## 5. Conclusions

Our study demonstrated that sucrose or sucralose increased T1R2 and T1R3 taste receptors than in turn stimulated intestine and circulating levels of GIP and GLP-1, leading to hyperinsulinemia. Furthermore, the combination of sucrose or sucralose with a high fat diet aggravated the insulin signaling pathway in white adipose tissue, decreasing the abundance of GLUT-4, resulting in hyperglycemia. These sweeteners with and without a high fat diet increased the adipocyte size and lipogenesis and stimulated the inflammatory pathways mediated by TLR4. Insulin resistance produced by the consumption of sucrose or sucralose was associated with high levels of BCAA. The present work demonstrated that honey has a protective effect on the functionality of the adipocyte, reducing the inflammatory state probably due to the presence of several antioxidant compounds, the variety of monosaccharides and the low sucrose content. Therefore, this study indicates that consumers should be aware of the type of sweeteners and the content of fat in the diet that can produce metabolic abnormalities associated with the functionality of the adipose tissue. 

## Figures and Tables

**Figure 1 nutrients-11-00880-f001:**
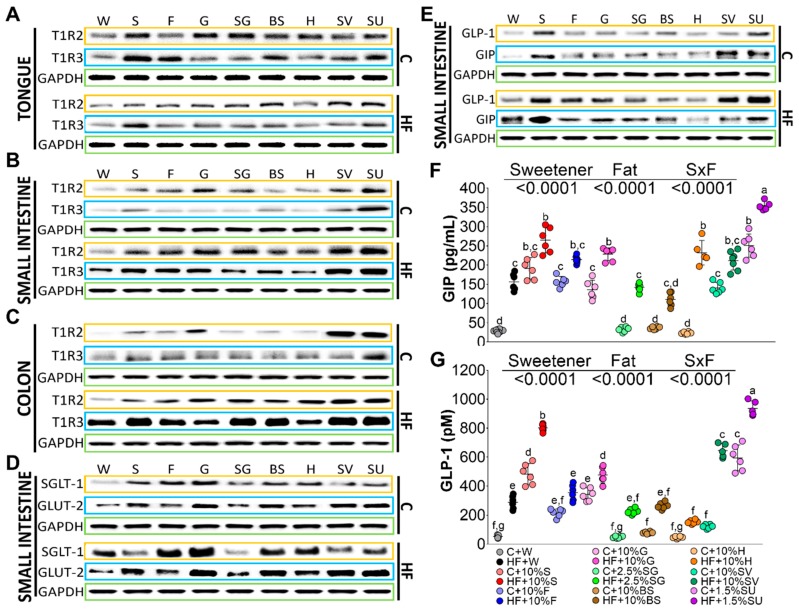
Taste receptors TIR2 and T1R3 in tongue (**A**), small intestine (**B**) and colon (**C**); glucose transporters (**D**) and incretins (**E**) in small intestine. Serum levels of glucose insulinotropic peptide (GIP) (**F**) and glucagon like peptide (GLP-1) (**G**) of rats fed with different sweeteners with or without high fat diet. The data are expressed as the mean ± SEM (*n* = 6). Results were considered statistically significant at *p* < 0.05. Significant differences among groups are indicated by letters, where a > b > c > d > e > f > g.

**Figure 2 nutrients-11-00880-f002:**
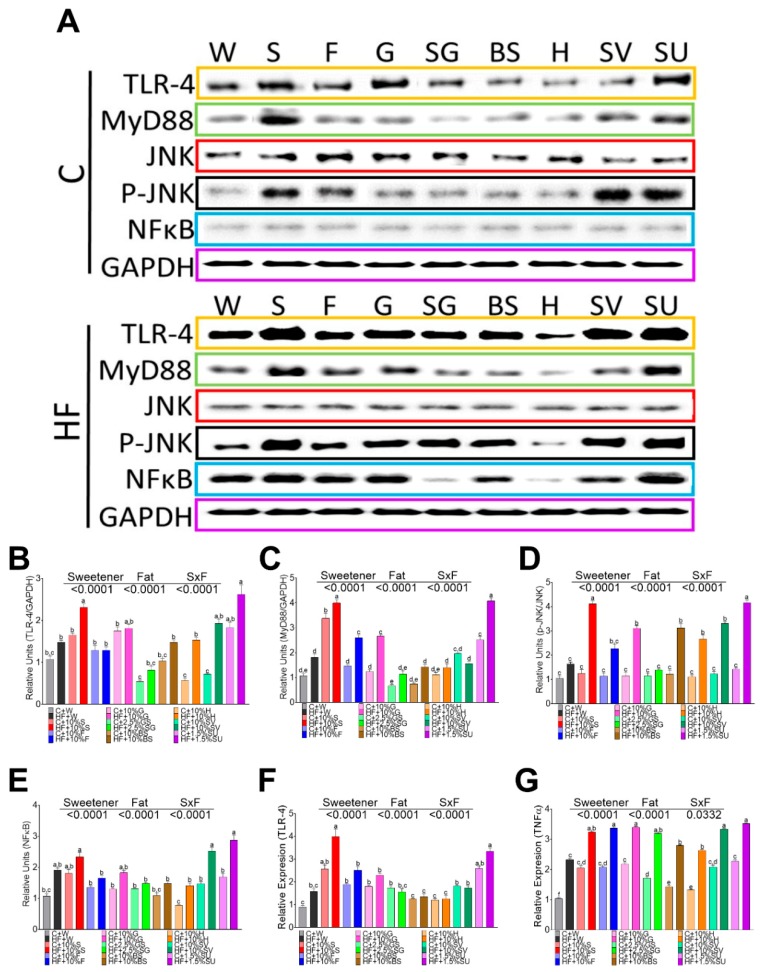
Relative protein abundance and gene expression of inflammation markers in white adipose tissue of rats fed different sweeteners with and without high fat diet (**A**); protein abundance of toll like receptor 4 (TLR-4) (**B**), myeloid differentiation factor 88 (MyD88) (**C**); c-Jun N- terminal kinase (JNK) (**D**); Nuclear Factor kappa-light-chain-enhancer of activated B cells (NFκB) (**E**); Relative expression of TLR4 (**F**) and tumor necrosis factor alpha (TNFα)(**G**). The data are expressed as the mean ± SEM (*n* = 6). Results were considered statistically significant at *p* < 0.05. Significant differences among groups are indicated by letters, where a > b > c > d > e.

**Figure 3 nutrients-11-00880-f003:**
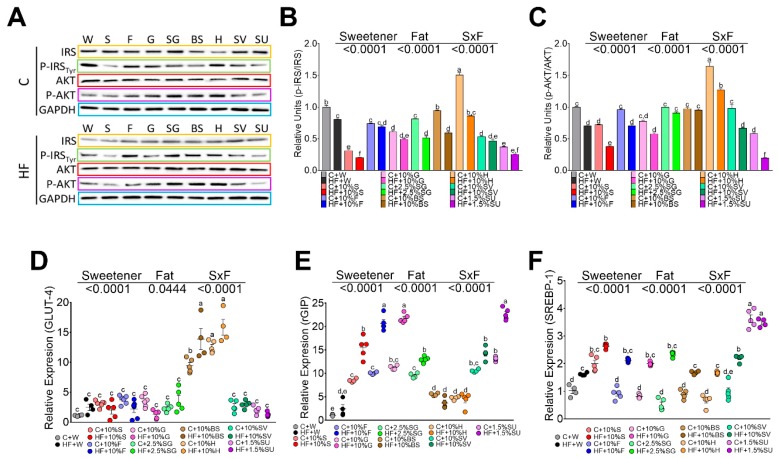
Relative protein abundance of insulin signaling pathway in white adipose tissue. IRS, pT263 ^IRS1^, AKT, and pS473^AKT^ in rats fed different sweeteners with or without high fat diet (**A**); phosphorylated IRS/total IRS ratio (**B**); phosphorylated AKT/total AKT ratio (**C**); relative expression of glucose transporter type 4 (GLUT 4) (**D**), GIP receptor (**E**) and sterol regulatory element-binding protein 1 (**F**). The data are expressed as the mean ± SEM (*n* = 6). Results were considered statistically significant at *p* < 0.05. Significant differences among groups are indicated by letters, where a > b > c > d > e.

**Figure 4 nutrients-11-00880-f004:**
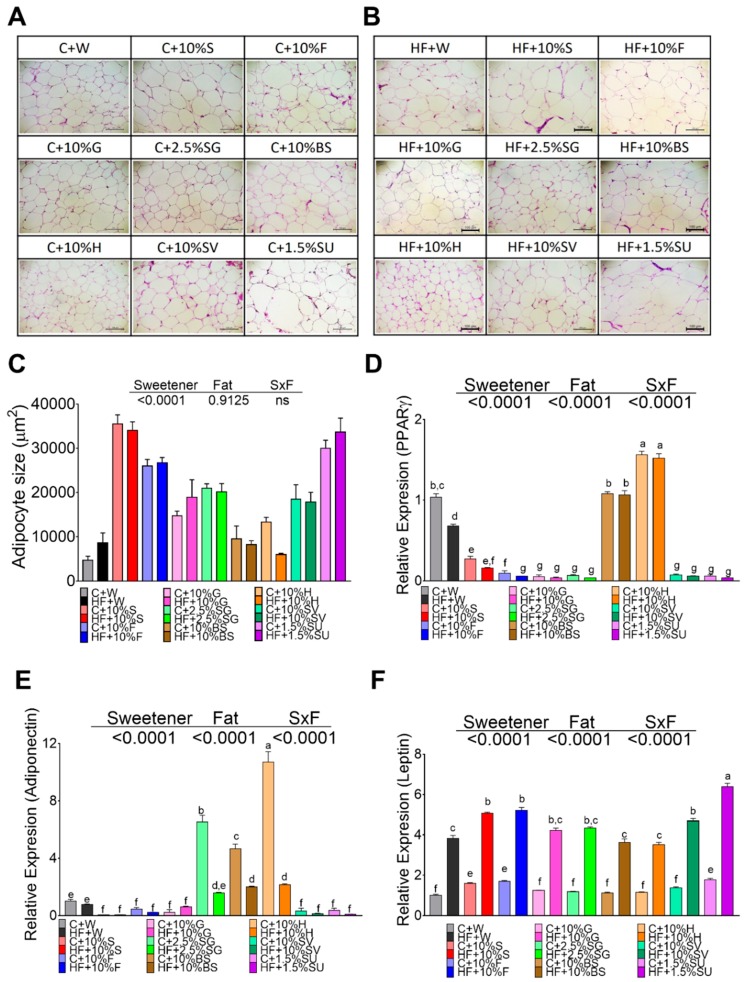
The type of sweetener and high fat diet differentially modify the functionality of the adipocyte in white adipose tissue. Hematoxylin-Eosin staining (HE) of white adipose tissue of rats fed different sweeteners with control (**A**) or high fat diet (**B**); adipocyte size (**C**); mRNA relative expression of peroxisome proliferator-activated receptor gamma (PPARγ) (**D**), adiponectin (**E**) and leptin (**F**). The data are expressed as the mean ± SEM (*n* = 6). Results were considered statistically significant at *p* < 0.05. Significant differences among groups are indicated by letters, where a > b > c > d > e > f.

**Figure 5 nutrients-11-00880-f005:**
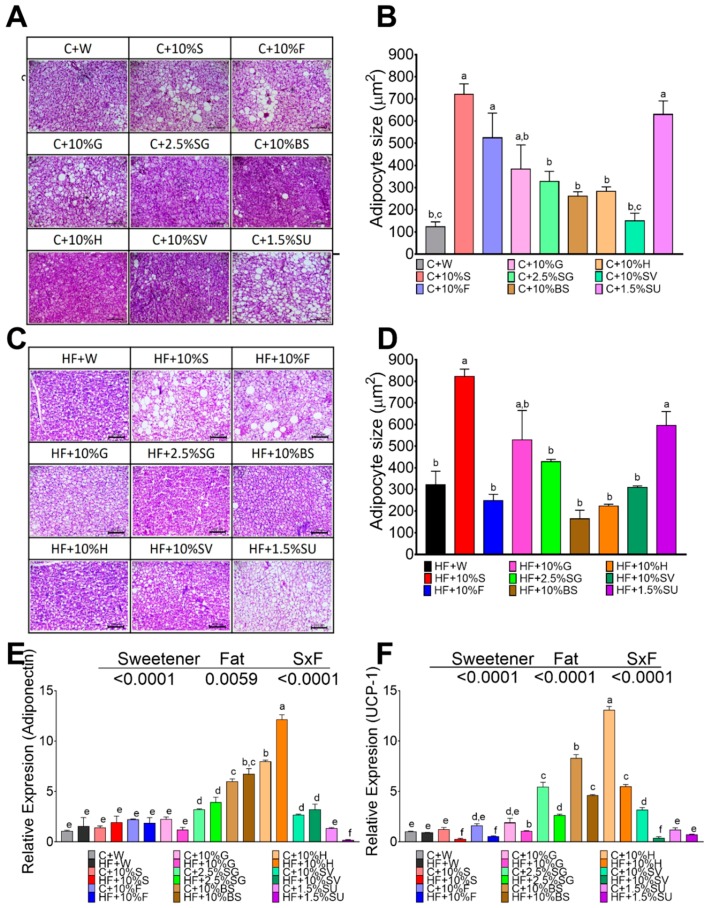
Effect of type of sweeteners on histological morphology and expression of adiponectin and UCP-1 in brown adipose tissue. Hematoxylin-Eosin staining (HE) of BAT of rats fed different sweetener with control (**A**) or high fat diet (**C**). Brown adipocyte size (**B**,**D**); Relative expression of adiponectin (**E**) and uncoupling protein 1 (UCP-1) (**F**). The data are expressed as the mean ± SEM (*n* = 6). Results were considered statistically significant at *p* < 0.05. Significant differences among groups groups are indicated by letters, where a > b > c > d > e > f.

**Figure 6 nutrients-11-00880-f006:**
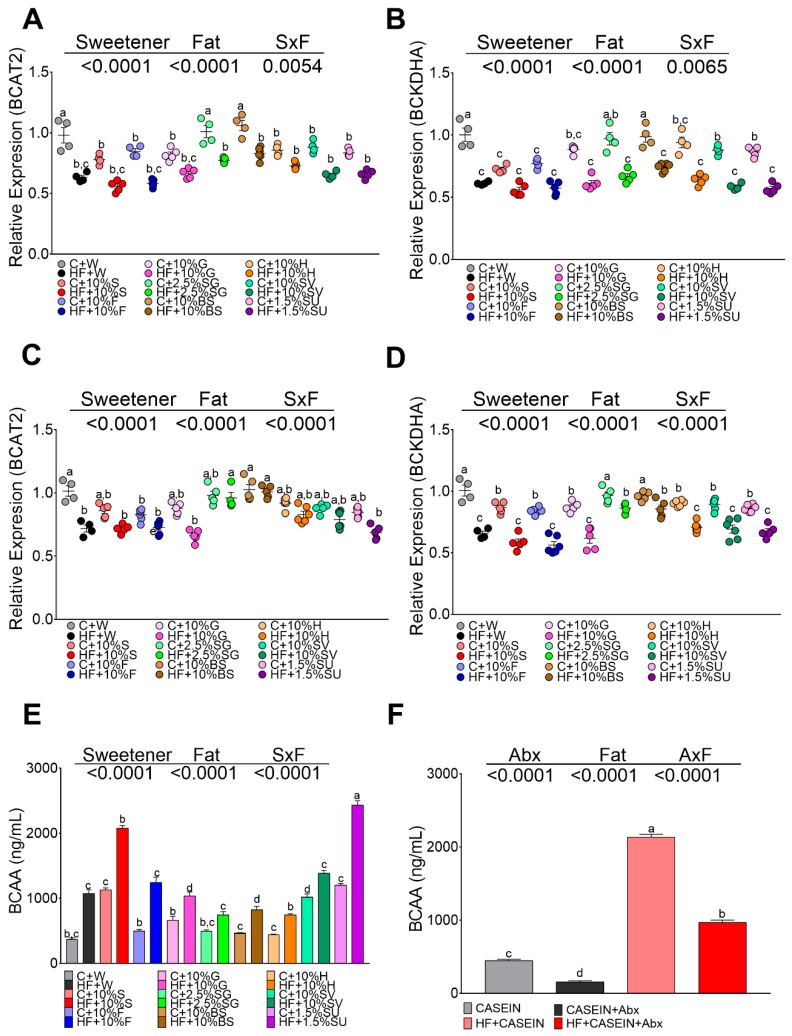
mRNA abundance of amino acid catabolizing enzymes in white adipose tissue (WAT) and brown adipose tissue (BAT), and circulating branched chain amino acids (BCAA) in rats fed different sweeteners with and without high fat diet. mRNA abundance of branched chain aminotransferase-2 (BCAT2) (**A**) and branched chain alpha keto acid dehydrogenase (BCKDH) (**B**) in WAT and BAT (**C**,**D**). Serum BCAA (valine, isoleucine, leucine) (**E**). Effect of antibiotics on serum BCAA (**F**). The data are expressed as the mean ± SEM (*n* = 6). Results were considered statistically significant at *p* < 0.05. Significant differences among groups groups are indicated by letters, where a > b > c > d > e.

**Table 1 nutrients-11-00880-t001:** Body weight and serum biochemical parameters of rats fed natural or artificial sweeteners with and without high fat.

Group	Body Weight (g)	Glucose (mg/dL)	Insulin (ng/mL)	Leptin (ng/mL)
C + W	546.4 ± 12.6 ^c^	79.19 ± 3.3 ^c^	0.24 ± 0.05 ^h^	3.92 ± 0.08 ^g^
HF + W	628.2 ± 16 ^a^	93.55 ± 3.7 ^b, c^	2.40 ± 0.17 ^g^	17.58 ± 0.24 ^d^
C + 10%S	680 ± 16.33 ^a^	143.5 ± 3.5 ^a,b^	5.00 ± 0.21 ^e^	9.48 ± 0.43 ^e^
HF + 10%S	720.2 ± 3.5 ^a^	174.8 ± 7.1 ^a^	6.78 ± 0.24 ^b^	35.62 ± 1.17 ^b^
C + 10%F	558.5 ± 1 ^b^	115.4 ± 6.2 ^b^	2.83 ± 5.4 ^g^	6.92 ± 0.25 ^f^
HF + 10%F	587.2 ± 18.7 ^b^	132.8 ± 2.4 ^a, b^	5.39 ± 0.16 ^d^	15.81 ± 0.3 ^d^
C + 10%G	589.7 ± 22.3 ^b^	133 ± 5.7 ^a, b^	1.72 ± 0.08 ^g^	4.94 ± 0.15 ^g^
HF + 10%G	606.8 ± 27.3 ^a, b^	137.9 ± 4.5 ^a, b^	3.97 ± 0.05 ^f^	22.78 ± 0.62 ^c^
C + 2.5%SG	562.2 ± 17.5 ^c^	89.18 ± 2.6 ^b, c^	0.28 ± 0.04 ^h^	5.64 ± 0.26 ^e^
HF + 2.5%SG	540 ± 12.05 ^c^	138.6 ± 6.9 ^a, b^	1.72 ± 0.15 ^g^	9.48 ± 0.37 ^e^
C + 10%BS	593.2 ± 6.5 ^b^	108.4 ± 3.3 ^b^	0.71 ± 0.09 ^h^	3.63 ± 0.27 ^f^
HF + 10%BS	599.2 ± 10.73 ^a, b^	122.2 ± 3.3 ^b^	1.76 ± 0.12 ^g^	7.75 ± 0.26 ^e^
C + 10%H	574.3 ± 18.11 ^b^	85.7 ± 4.7 ^b, c^	0.31 ± 0.03 ^h^	3.62 ± 0.19 ^g^
HF + 10%H	591 ± 11.79 ^b^	84.8 ± 4.7 ^b, c^	0.61 ± 0.08 ^h^	9.57 ± 0.29 ^e^
C + 10%SV	609 ± 19.26 ^a, b^	138.7 ± 8.1 ^a, b^	3.7 ± 0.11 ^f^	8.41 ± 0.35 ^e^
HF + 10%SV	634.2 ± 19.29 ^a^	141.4 ± 3.4 ^a, b^	5.05 ± 0.20 ^e^	27.21 ± 0.52 ^c^
C + 1.5%SU	603.3 ± 21.74 ^a, b^	151.5 ± 2.7 ^a^	6.44 ± 0.35 ^c^	25.54 ± 1.32 ^c^
HF + 1.5%SU	540.8 ± 10.4 ^c^	177 ± 6.5 ^a^	10.6 ± 0.31 ^a^	39.80 ± 0.74 ^a^
Sweetener	<0.001	<0.001	<0.001	<0.001
Fat	0.0349	<0.001	<0.001	<0.001
p	0.0036	<0.001	<0.001	<0.001

C, control diet; HF, high fat diet; W, water; S, sucrose; F, fructose; G, glucose; SG, steviol glycosides; BS. Brown sugar; H, honey; SV, SV; SU, sucralose. Different letter superscripts indicate significant differences among groups (a > b > c > d > e > f > g > h).

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
