# Peer review of "Natural and Artificial Sweeteners and High Fat Diet Modify Differential Taste Receptors, Insulin, and TLR4-Mediated Inflammatory Pathways in Adipose Tissues of Rats"

_nutrients, 2019, doi:10.3390/nu11040880_

Round 1
Reviewer 1 Report
It is not known whether sweetener, high fat (HF) or combination is responsible for obesity. The purpose of the present work was to study the effects of different types of sweeteners on the functionality of the adipocyte. The authors found that taste receptors T1R2 and T1R3 were differentially expressed in tongue and intestine by sweeteners and HF. The combination of fat and sweetener showed an additive effect on circulating levels of GIP and GLP-1. In adipose tissue, sucrose and sucralose stimulated TLR4,
and JNK. The combination of a HF with sweeteners increased NFB with the exception of SG and honey. Honey maintained active the insulin signaling pathway and the smallest adipocytes in WAT and BAT and the highest expression of adiponectin, PPAR and UCP-1 in BAT. Based on the data, the authorssuggest that consumption of honey or brown sugar maintained functional adipocytes despite the consumption of a HF diet.
The aim of this study to ask “whether sweetener, high fat (HF) or combination is responsible for obesity”is of importance. However, the study deals with so many different sugars/sweeteners and analyzed their effects on many parameters, tongue, intestine, adipose tissue, incretins, cytokines and insulin signaling, in a manner quite scattered without clear logical connection between them. Hence it is quite difficult to read. The authors should more clearly and simply state what is new and relevant. Discussion should be focused to them.
Comments
1. To me, the broad beneficial effect of honey, which is distinct form most of other sugars/sweeteners, is the novel and important finding of this study. This should be more clearly stated and more deeply discussed.
2. Manuscript is rather descriptive and lacks mechanistic analysis and discussion.
3. How the changes in GLP-1 and GIP are related to the functions of WAT and BAT, intestine or tongue should be shown or discussed.
4. Initial par of Discussion is repetitive and redundant with Introduction and should be largely reduced or omitted.
5. First paragraph of Discussion contains too many issues and is too long. Second paragraph is similar. The paragraph should be divided into a few, in which each paragraph has a focused issue, such as, for instance, new finding of the present study, the scientific relevance, nutritional/clinical message, or comparison with previous reports.
Author Response
Response to the reviewers.
We appreciated the time spent in the revision of the manuscript, your suggestions and comments.
The suggestions or comments from the Editors and Reviewers were taken into account and are described below.
English language and minor spell were checked
Reviewer 1.
1. Thank you very much for your suggestion, we modified the discussion to give more importance to the beneficial effects of honey.
2. We modified the discussion section to show a mechanistic analysis
3. We included in the introduction section the role of GLP and GIP in WAT and BAT, also in the discussion section we describe the potential mechanism for which GIP regulates the insulin pathway
4. We deleted the first paragraph of the discussion to avoid repetition of the introduction.
We modified the discussion section into short paragraphs and we give more emphasis to the new findings and relevance of the honey
Reviewer 2 Report
Manuscript ID: nutrients-473031
Title: Natural and artificial sweeteners and high fat diet modify differentially taste receptors, insulin and TLR4-mediated inflammatory pathways in adipose tissue of rats
Authors: Nimbe Torres, Mónica Sánchez-Tapia, Jonathan Martínez-Medina, Armando R Tovar
General comment
In this paper, Authors studied the effects of feeding of rats with different types of natural and artificial sweeteners without/with high fat diet on the expression levels of sweet taste receptors, glucose transporters and incretins. Authors also studied the effects of those sweeteners on the adipocyte functions such as inflammation and insulin signaling pathways, adipokines expression in WAT, and UCP-1 expression in BAT as well as the adipocyte size. Since the choices of the sweeteners are rather wide and systematic, the data are interesting and significantly contribute to this research field. However, there are several problems in the description and interpretation of the data. For the reasons above, this manuscript seems to merit publication although some revision is necessary before the manuscript is accepted for publication in the Nutrients journal.
Specific points
1. Some important references on the role of sweet taste receptor in adipocyte function are not cited in the Introduction section, e.g. Masubuchi Y, et al. PLoS One. 2013; 8(1): e54500; Masubuchi Y, et al. PLoS One. 2017 May 4; 12(5): e0176841.
2. Line. 38: sucralose is not a natural sweetener.
3. Table 1: Author should describe what the abbreviations (C, W, HF, S, F etc…) mean for.
4. There are several problems in the description and interpretation of the data.
Figure 1.
Line137-138: the words, fructose, brown sugar sucralose appear twice in the same sentense, that does not make sense.
Line 156: the inducing effect of SV on GLP-1 is weak and seems comparable to those of F, G, SG, BS.
Line 157: The effects on GLP-1 of F and SG are also enhanced by high fat.
Figure 2.
Line 174: The effect of glucose on TLR4 is comparable to that with sucrose or sucralose.
In Paragraph 3.5., there are several inconsistent and wrong description of the data.
5. In Figure 2, the NFkB expression levels do not correlate with upstream signals such as TLR4 and p-JNK. The activation level of NFkB should be examined, e.g. phosphorylation or nuclear localization of NFkB.
6. The legend is missing for Figure 6.
Author Response
Reviewer 2
1. We added the references suggested by the reviewer on the role of sweet receptor on adipocytes
2. Line 38 and 58. You are correct. We added the word artificial
3. Abbreviations of the sweeteners were added to the text and table 1
4. Line 137-138. We double checked the sentence and we modified the description of the results and added the percentages of increase in the abundance of T1R2, T1R3, GIP and GLP-1 with respect to control group (C+W) to clarify the description of the results.
5. Line 157. We modified the text, in effect , the TLR4 abundance was similar for sucrose, glucose and sucralose groups
6. Sweeteners can not differentially affect several pathways at the same time. Unfortunately we did not measure phosphorylation of NFkB, however this transcription factor regulate the production of inflammatory cytokines particularly TNFalpha. As you can see in Fig 2G, there was an increase in TNFalpha with some sweeteners, interestingly the addition of high fat in the diet significantly increased the gene expression of TNFalpha with all sweeteners at different extent, these results were associated with an increase in TLR4, JNK and total NFkB abundance.
7. We added the figure legend to Figure 6.
Sincerely,
Nimbe Torres PhD
Round 2
Reviewer 1 Report
The authors reasonably respond to comments of this reviewer.